# Psychological safety in European medical students' last supervised patient encounter: A cross-sectional survey

**Cathinka Thyness**⊙*, **Hilde Grimstad, Aslak Steinsbekk**⊙

Department of Public Health and Nursing, Norwegian University of Science and Technology, Trondheim, Norway

\* cathinka.thyness@ntnu.no

## Abstract

### Objective

To investigate the association between European medical students' psychological safety in and experiences from their last supervised patient encounter.

### Materials and methods

A cross-sectional online survey among European medical students. Bivariable and multivariable linear regression was used to explore the associations between the dependent variable psychological safety and independent variables concerning students' experiences from their last supervised patient encounter.

### Results

A total of 886 students from more than 25 countries participated. The variables most strongly associated with psychological safety were supervisor coaching and modelling behaviour, adjusted beta 0.4 (95%CI 0.3 to 0.5) and 0.1 (95%CI 0.1 to 0.2) per unit respectively on a one-to-five-point scale, and studying in Northern Europe, adjusted beta 0.4–0.5 compared to other regions. There was a weak negative association (reduced score on psychological safety) for being supervised by a medical doctor with <5 years' experience and a positive association for student confidence. Student gender, student seniority, speciality, whether peers were present, number of previous encounters with the supervisor and supervisor articulation and exploration behaviour were not associated in multivariable analysis.

### Conclusion

Coaching might be a good primary focus to improve supervision practices, as participation with feedback is known to be beneficial for learning and coaching was strongly associated with psychological safety. Supervisors in western, eastern, and southern Europe might have to work harder to create psychological safety than their northern colleagues.

**Data Availability Statement:** All relevant data are within the paper and its Supporting Information files.

**Funding:** The author(s) received no specific funding for this work.

**Competing interests:** The authors have declared that no competing interests exist.

## Introduction

Psychological safety is the belief that you will not be punished or humiliated for speaking up with ideas, questions, concerns, or mistakes in a group [1]. In a psychologically safe environment, it is appreciated when people are open and honest about their lack of knowledge or skills or share concerns about how things are done as these are recognised as learning opportunities [1].

In the context of education, psychological safety concerns the psychosocial learning environment. The learning environment as perceived by students, i.e. the learning climate, has been recognised as important to learning for decades [2]. A meta-analysis showed that higher psychological safety in non-medical education predicts more learning behaviour and better performance [3]. Psychological safety appears to be especially important in contexts where there is complexity and social impact [4], such as in health care [1].

Qualitative studies have found that medical students talk of issues related to psychological safety as promoting their self-directed learning [5], professionalism [6] and (pro)activity [7, 8]. In medical students' supervised encounters with patients, where education meets health care, there was an association between the students' perception of learning and psychological safety [9].

This indicates that promoting psychological safety in medical students' supervised encounters with patients is important. However, we have found no studies looking at the association between psychological safety and medical students' experiences with other aspects of supervised patient encounters. Such knowledge can be helpful in unravelling what can be done to promote psychological safety in this setting.

Our aim was therefore to investigate the association between European medical students' psychological safety in and experiences from their last supervised patient encounter.

## Materials and methods

### Design and ethics

This was a cross-sectional survey where European medical students were invited to complete an online questionnaire. It is part of a larger study on European medical students' self-reported experience from supervised patient encounters.

Data were collected anonymously online and without questions about the participants health. All handling of data and analysis was conducted in Norway. After consultation with The Norwegian Centre for Research Data (reference number 235211), the study was deemed exempt from ethical and privacy approval. Participants received written information about the study and contact information for the study on the first page of the survey and a note before the submit button that by clicking "send" they agreed to participate.

### Participants, recruitment, and data collection

Inclusion criteria were (1) being a medical student, (2) studying in Europe, (3) having seen patients as part of the medical school curriculum, (4) having had at least one experience of a supervised patient encounter and (5) having completed the psychological safety scale. Eligibility was assessed based on self-reported answers to questions in the survey.

To recruit participants, the online survey was distributed as web-links in e-mails and on social media by student representatives in national medical student associations in the International Federation of Medical Students Associations (IFMSA) between March and October 2020. Individual students were also encouraged to share the link with their peers.

Data were collected through the secure-server tool www.nettskjema.no. The questionnaire was in English only to ensure uniformity, as it was expected that European medical students would be capable of comprehending and responding in English.

## Questionnaire and variables

Students were asked to respond to the questions about supervision based on the last time they were with a patient and received supervision as part of their medical school curriculum.

**Dependent variable–psychological safety.** Edmondson's psychological safety scale [10] was used with minor modifications to fit the situation of clinical supervision of medical students (e.g. "this team" was changed to "those present") (S1 File). The scale has been used in a variety of settings including health care [3] and adaptations are often made [11–13]. Respondents rated each of the 7 statements on a 7-point Likert scale ranging from "very inaccurate" (1) through "neither inaccurate nor accurate" (4) to "very accurate" (7). Responses to reverse statements were recoded before a mean value was calculated with 1 being the lowest and 7 the highest psychological safety score. In this study, Cronbach's alpha was 0.72.

**Independent variables.** Students reported on their age and gender which was categorised as male or not male (female, non-binary, do not wish to specify). In post-graduate medical education, research suggest gender impacts trainees' perception of psychological safety [14]). Male was chosen as the comparator, as some studies suggest males feel more psychologically safe [14].

Respondents were asked which country they studied in, and place of study was categorised according to the United Nations (UN) M49 classification of geographic regions [15]. They were also asked to report on the speciality in which supervision took place as research in post-graduate medical education has found differences in psychological safety between specialities [14, 16].

Furthermore, respondents were asked what year of medical school they were in and a variable was created where those in the last two years of their medical school were categorised as senior medical students. They were asked about number of students present which categorised as the student alone or with peers present, supervisor profession and level of experience which was categorised as doctor with number of years' experience or as other profession, and on how many previous occasions the supervisor had supervised the student. Supervisor behaviour was measured with the Maastricht Clinical Teaching Questionnaire (MCTQ), which has been found to be valid and reliable to evaluate clinical supervisors [17]. It contains 14 statements grouped into 5 sub-scales of which the sub-scales modelling, coaching, articulation and exploration were entered into the analyses as continuous variables. These subscales had Cronbach's alphas ranging from 0.74 to 0.87 in this study. These variable were chosen as qualitative studies in undergraduate medical education have suggested that student seniority [7], having peers present [8], supervisor seniority [7], having the same supervisor over time [8, 18–21] and supervisor behaviour [8] impacts factors related to psychological safety.

Students' own judgement of their confidence was measured with the statement "I am able to express myself and show confidence" taken from the Clinical Learning Evaluation Questionnaire [22], as research in post-graduate medical education suggests trainees' confidence impacts on perception of psychological safety [23].

## Analyses

Descriptive statistics were used to describe the sample. Associations between psychological safety and the input variables were tested with bivariable and multivariable linear regression. Any input variable with a p-value <0.2 in bivariable linear regression was included in the multivariable regression model. Gender was included as a control variable regardless of p-value.

Beta-coefficients (B) and 95% confidence intervals (95% CI) were used to describe the results. Data were analysed using Stata version 17 (StataCorp, College Station, Texas, USA).

## Results

### Respondents

Of the 923 questionnaires received, 886 fulfilled inclusion criteria and were thus included in the analysis (Fig 1).

Among the respondents, 27% were male and the median age was 24. Respondents studied in more than 25 countries from all four European regions (Table 1). Nearly all were enrolled in six-year programmes, with about half in the last two years of their programme. The supervision took place in over 16 different specialities with most responses from internal medicine (n = 243) and paediatrics (n = 89).

### Association with psychological safety

Fig 2 displays the psychological safety scores of the participants. The mean psychological safety score on the one to seven scale was 4.9 (standard deviation 1.0, Range 1.4 to 7.0).

All variables in Table 2 except male gender had a p-value <0.2 in the bivariable regression and were therefore included in the multivariable regression analysis.

In the multivariable regression analysis (Table 2), 5 out of the 12 independent variables were associated with psychological safety. For each one-point increase in the MCTQ Coaching scale which ranges from 1 to 5, the psychological safety score increased with 0.4 while for MCTQ modelling the increase was 0.1. Studying in the Northern European region increased the psychological safety score with 0.4 to 0.5 points compared to the other regions. For the other independent variables, there were no changes above 0.2 points, although there was a negative association (reduced score on psychological safety) for being supervised by a medical doctor with <5 years' experience and a positive association for student confidence.

## Discussion

Psychological safety was most strongly associated with scoring supervisor coaching and modelling behaviour highly and studying in Northern Europe in the multivariable analysis. There was a weak negative association (reduced score on psychological safety) for being supervised by a medical doctor with <5 years' experience and a weak positive association for student confidence.

The psychological safety scores ranged from 1.4 to 7.0, indicating variation in how psychological safe the students felt in their latest supervised patient encounter. Such variation shows that there are supervised encounters that can be learned from and others that could benefit from improvements. To the best of our knowledge, no cut-off point exists regarding which

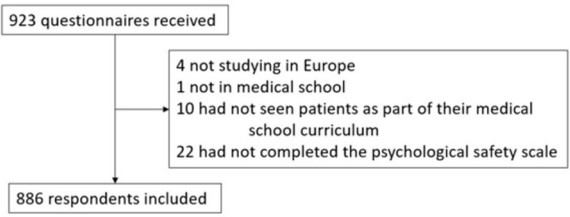

**Fig 1. Flow chart.**

**Table 1. Characteristics of the sample (N = 886).**

| Characteristic | Number of respondents (%) | Psychological safety score Mean (SD) |
|---|---|---|
| **All respondents** | **886 (100%)** | **4.9 (1.0)** |
| Male | 239 (27%) | 5.0 (1.0) |
| • Not male | 646 (73%) | 4.9 (0.9) |
| Age | | |
| • 18–21 | 148 (17%) | 4.8 (2.7) |
| • 22–24 | 396 (46%) | 4.9 (1.0) |
| • 25–27 | 224 (26%) | 5.0 (0.9) |
| • >27 | 96 (11%) | 5.1 (1.0) |
| Country | | |
| • Eastern Europe | 266 (30%) | 4.8 (0.9) |
| • Bulgaria | 1 (<1%) | 5.3 (NA) |
| • Czech Republic | 17 (2%) | 5.0 (0.8) |
| • Hungary | 64 (7%) | 4.9 (0.9) |
| • Poland | 101 (11%) | 5.0 (0.9) |
| • Romania | 19 (2%) | 4.6 (0.8) |
| • Slovakia | 37 (4%) | 4.5 (0.9) |
| • Ukraine | 27 (3%) | 3.9 (0.6) |
| • Northern Europe | 286 (32%) | 5.4 (0.9) |
| • Denmark | 42 (5%) | 5.4 (0.9) |
| • Iceland | 7 (1%) | 5.2 (0.5) |
| • Latvia | 10 (1%) | 4.2 (1.0) |
| • Norway | 222 (25%) | 5.4 (0.8) |
| • Sweden | 2 (<1%) | 5.3 (0.8) |
| • United Kingdom | 3 (<1%) | 5.2 (0.5) |
| • Southern Europe | 107 (12%) | 4.6 (0.9) |
| • Bosnia and Herzegovina | 14 (2%) | 4.7 (1.0) |
| • Croatia | 54 (6%) | 4.7 (0.9) |
| • Greece | 28 (3%) | 4.4 (1.0) |
| • Malta | 7 (1%) | 4.1 (0.4) |
| • Portugal | 1 (<1%) | 4.6 (NA) |
| • Slovenia | 1 (<1%) | 4.7 (NA) |
| • Spain | 2 (<1%) | 4.3 (0.6) |
| • Western Europe | 187 (21%) | 4.7 (0.9) |
| • Belgium | 47 (5%) | 4.7 (1.0) |
| • France | 136 (15%) | 4.7 (0.8) |
| • the Netherlands | 1 (<1%) | 5.6 (NA) |
| • Switzerland | 3 (<1%) | 5.8 (0.1) |
| • Other | 38 (4%) | 4.9 (1.1) |
| • Cyprus | 23 (3%) | 4.6 (1.0) |
| • Other European country | 15 (2%) | 5.2 (1.0) |
| Speciality | | |
| • Family Medicine | 72 (8%) | 5.4 (0.9) |
| • Internal Medicine | 243 (28%) | 4.9 (1.0) |
| • Neurology | 72 (8%) | 5.0 (0.8) |
| • Obstetrics and Gynaecology | 44 (5%) | 4.8 (1.1) |
| • Orthopaedics | 43 (5%) | 5.0 (1.1) |
| • Paediatrics | 89 (10%) | 5.0 (1.0) |

(*Continued*)

**Table 1.** (Continued)

| Characteristic | Number of respondents (%) | Psychological safety score Mean (SD) |
|---|---|---|
| • Surgery | 70 (8%) | 4.7 (0.9) |
| • Other[1] | 245 (28%) | 4.9 (0.9) |
| Duration of medical school programme | | |
| • 2 years | 6 (1%) | 4.2 (0.1) |
| • 3 years | 7 (1%) | 4.9 (0.9) |
| • 4 years | 19 (2%) | 4.4 (0.7) |
| • 5 years | 21 (2%) | 4.7 (0.8) |
| • 6 years | 789 (90%) | 5.0 (1.0) |
| • 7 years | 34 (4%) | 4.7 (0.7) |
| In last two years of medical school | 460 (53%) | 4.9 (1.0) |
| • Not last two years | 415 (47%) | 5.0 (0.9) |
| Supervised with at least one other student | 579 (68%) | 4.9 (1.0) |
| • Supervised alone | 275 (32%) | 5.0 (1.0) |
| Supervisor profession and seniority | | |
| • Doctor with >10 years' experience | 382 (43%) | 5.0 (1.0) |
| • Doctor with 5–10 years' experience | 191 (22%) | 4.9 (0.9) |
| • Doctor with <5 years' experience | 193 (22%) | 4.8 (0.9) |
| • Doctor, unknown experience | 101 (11%) | 5.0 (1.0) |
| • Other | 12 (1%) | 4.7 (0.7) |
| How many times before the supervisor had supervised the respondent | | |
| • 0 | 324 (38%) | 4.9 (1.0) |
| • 1–5 | 343 (41%) | 5.0 (0.9) |
| • 5–10 | 82 (10%) | 5.1 (0.9) |
| • >10 | 93 (11%) | 5.0 (1.0) |
| Agreed that "I am able to express myself and show confidence" | 552 (62%) | 5.1 (0.9) |
| • Disagreed | 334 (38%) | 4.6 (0.9) |

Notes: 1 Including Anaesthesia, dermatology, ear, nose and throat, oncology, ophthalmology, palliative care, psychiatry, radiology, and rehabilitation.

Abbreviations: SD = standard deviation, NA = not applicable,

level of score on the 1–7 scale constitutes a psychologically safe environment. In Hommes et al.'s study, the subset of students interviewed from a year group that scored psychological safety in tutorial groups to about 5.3, said they felt psychologically unsafe in these tutorial groups [24]. In our current study, students from Northern Europe had the highest psychological safety score (5.4). Still, in a qualitative study we conducted in Northern Europe (Norway and England), students expressed they often felt unsafe in supervision of patient encounters and that this hindered their learning [8]. Psychological safety cannot be too high [1], but what is sufficient to ensure student engagement in learning is uncertain and probably depends on individual differences in confidence [8, 23, 25].

It is well known that leadership matters to psychological safety [1, 3, 4]. This indicates that supervisors are likely to influence students experience of psychological safety. This was supported by the finding that all the four measured supervisor behaviours were positively associated with psychological safety in bivariable analysis. Among these, coaching and modelling were still associated with psychological safety when controlling for other variables. Coaching had the strongest association with psychological safety, with an estimated 1.6-point increase on a 7-point scale for psychological safety between students that rated coaching the lowest and highest. Coaching was based on students' ratings of supervisors' teaching to the student's level

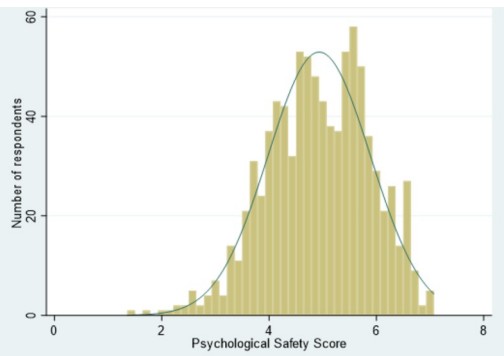

**Fig 2. Psychological safety scores (range one to seven) in the latest supervised patient encounter among European medical students with normal curve (n = 886).**

of experience, offering sufficient opportunities for independent practice and providing useful feedback. As this is a cross-sectional study, we do not know the direction of this relationship. Studies testing interventions to improve psychological safety in supervisor-student encounters is therefore needed. Considering that the strongest association was between psychological safety and coaching, and that practice with feedback, the key process of coaching, is a well-known driver of learning [26, 27], coaching might be a good primary focus to improve super-vision practices.

It was a clear finding that those studying in Northern Europe (a group dominated by those studying in Norway) scored higher on psychological safety than the rest of Europe. The other European regions were comparable to each other. The Nordic countries, and especially Scandi-navia (Sweden, Norway, Denmark), stand out internationally by their citizens having high lev-els of trust in people and institutions [28]. Kahn found that trusting relationships fostered psychological safety [29], and later studies have also found associations between trust and psy-chological safety [25]. It is therefore possible, that the differences in interpersonal trust between European countries can explain some of the geographical variance in psychological safety observed in this study. This suggests that supervisors in countries with lower levels of interpersonal trust might have to work harder than their Scandinavian colleagues to create psychological safety when supervising medical students.

## Strengths and weaknesses

With over 850 respondents we had the opportunity to examine the association between psy-chological safety and a considerable number of factors. The data were from a cross-sectional survey. So, no conclusions about cause and effect can be drawn. However, our findings can be used as inspiration for what variables to include in longitudinal research to further explore what impacts psychological safety in clinical education. How the survey was distributed means we do not know how many students we reached or if there was a systematic difference between those who did and those who did not answer. We hope our description of the sample can help readers evaluate if the sample is relevant to their context. The survey was advertised to be about clinical supervision, so students should not have been primed to participate due to a par-ticularly striking experience of psychological safety or lack thereof. The survey was sent out shortly after the COVID-19 pandemic hit, which affected clinical learning activities. Students were asked to respond based on a situation where they were with a patient, which means the situation should be similar to non-covid clinical learning. Students might have responded

**Table 2. Regression analyses of associations between psychological safety and context, student characteristics, structural aspects of clinical supervision, supervisor behaviour, and student confidence.**

| Characteristics | Bivariable regression | Multivariable regression |
|---|---|---|
| | Crude B-coefficient (95% CI) | Adjusted B-coefficient (95% CI) |
| European region | | |
| • Northern | Reference | Reference |
| • Eastern | -0.6† (-0.8, -0.5) | -0.4*(-0.6, -0.3) |
| • Southern | -0.8† (-1.0, -0.6) | -0.5*(-0.7, -0.3) |
| • Western | -0.7† (-0.9, -0.5) | -0.4*(-0.6, -0.2) |
| • Other | -0.5† (-0.8, -0.2) | -0.5*(-0.8, -0.2) |
| Speciality | | |
| • Other | Reference | Reference |
| • Family Medicine | 0.5† (0.3, 0.8) | 0.1 (-0.1, 0.4) |
| • Internal Medicine | 0.1 (-0.1, 0.2) | 0.0 (-0.1, 0.2) |
| • Neurology | 0.1 (-0.2, 0.3) | -0.1 (-0.3, 0.1) |
| • OBGYN | 0.0 (-0.4, 0.3) | 0.0 (-0.2, 0.3) |
| • Orthopaedics | 0.2 (-0.2, 0.5) | -0.1 (-0.4, 0.1) |
| • Paediatrics | 0.1 (-0.1, 0.3) | 0.1 (-0.1, 0.3) |
| • Surgery | -0.2† (-0.4, 0.1) | 0.0 (-0.2, 0.2) |
| Senior student | -0.2† (-0.3, 0.0) | -0.1 (-0.2, 0.0) |
| Male student | 0.0 (-0.1, 0.2) | 0.1 (-0.0, 0.2) |
| Peers present | -0.1† (-0.2, 0.0) | 0.1 (0.0, 0.3) |
| Supervisor seniority | | |
| • MD >10 years' experience | Reference | Reference |
| • MD 5–10 years' experience | 0.0 (-0.2, 0.1) | 0.0 (-0.1, 0.1) |
| • MD <5 years' experience | -0.2† (-0.4, 0.0) | -0.2*(-0.3, 0.0) |
| • MD, unknown experience | 0.0 (-0.2, 0.2) | 0.1 (-0.1, 0.3) |
| • Other | -0.3 (-0.8, 0.3) | 0.0 (-0.4, 0.5) |
| Previous supervision encounters | | |
| • Never | Reference | Reference |
| • 1–5 | 0.1 (-0.1, 0.2) | 0.0 (-0.1, 0.1) |
| • 5–10 | 0.2† (0.0, 0.4) | 0.1 (-0.1, 0.3) |
| • >10 | 0.1 (-0.1, 0.3) | -0.1 (-0.3, 0.1) |
| MCTQ Modelling | 0.4† (0.4, 0.5) | 0.1* (0.1, 0.2) |
| MCTQ Coaching | 0.5† (0.5, 0.6) | 0.4* (0.3, 0.5) |
| MCTQ Articulation | 0.4† (0.3, 0.4) | 0.0 (-0.1, 0.1) |
| MCTQ Exploration | 0.2† (0.2, 0.3) | 0.0 (0.0, 0.1) |
| Agreed that "I am able to express myself and show confidence" | 0.5† (0.4, 0.6) | 0.1* (0.0, 0.3) |

Abbreviations: 95% CI = 95% confidence interval, OBGYN = Obstetrics and gynaecology, MCTQ = Maastricht Clinical Teaching Questionnaire

† p-value <0.2

*p-value <0.05

several weeks after suspension of clinical learning activities, though, which makes inaccurate recall a threat to the reliability of our findings.

## Supporting information

**S1 File.**
(DOCX)

**S1 Dataset.**
(DTA)

## Acknowledgments

We wish to thank medical student Vebjørn Andersson for his help with developing and distributing the survey, including facilitating the collaboration with the International Federation for Medical Student Association (IFMSA). Furthermore, IFMSA for their endorsement and all the student representatives that helped in distributing the survey, and students who answered the questionnaire.

## Author Contributions

**Conceptualization:** Cathinka Thyness, Hilde Grimstad.

**Data curation:** Cathinka Thyness.

**Formal analysis:** Cathinka Thyness, Hilde Grimstad.

**Investigation:** Cathinka Thyness.

**Methodology:** Cathinka Thyness, Hilde Grimstad.

**Project administration:** Cathinka Thyness.

**Supervision:** Hilde Grimstad.

**Visualization:** Cathinka Thyness.

**Writing – original draft:** Cathinka Thyness.

**Writing – review & editing:** Cathinka Thyness, Hilde Grimstad.

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
