## [Decision Letter · Decision Letter 0]

2 Mar 2023

PONE-D-22-23578Psychological safety in European medical students’ last supervised patient encounter: A cross-sectional surveyPLOS ONE

Dear Dr. Thyness, Thank you for submitting your manuscript to PLOS ONE. After careful consideration, we feel that it has merit but does not fully meet PLOS ONE’s publication criteria as it currently stands. Therefore, we invite you to submit a revised version of the manuscript that addresses the points raised during the review process.

We look forward to receiving your revised manuscript.

Kind regards,

Yaser Mohammed Al-Worafi

Academic Editor

PLOS ONE

Reviewers' comments:

Reviewer's Responses to Questions

**Comments to the Author**

1. Is the manuscript technically sound, and do the data support the conclusions?

Reviewer #1: Yes

Reviewer #2: Partly

2. Has the statistical analysis been performed appropriately and rigorously? 

Reviewer #1: Yes

Reviewer #2: Yes

3. Have the authors made all data underlying the findings in their manuscript fully available?

Reviewer #1: Yes

Reviewer #2: Yes

4. Is the manuscript presented in an intelligible fashion and written in standard English?

Reviewer #1: Yes

Reviewer #2: Yes

5. Review Comments to the Author

Reviewer #1: - Are there any impact/issues on psychological safety among medical students? If yes, please provide the evidence in order to show that this study is important

- Provide some explanation related to your variables.

- What method that the researcher used in selecting the participants?

- State the reliability result for Maastricht Clinical Teaching Questionnaire (MCTQ) in this study

- Under Questionnaire and variables, you mentioned that there are 12 independent variables were used in this study. Please what are the 12 independent variables.

- For the respondents, ‘not male’ refer to female or what? Please explain it

Reviewer #2: The authors investigated the relationship between supervisors’ attributes and behaviors and students’ psychological safety in clinical educational settings, an area of interest to readers. A sufficient number of data have been analyzed robustly quantitatively, and the methods are clear.

However, I have concerns regarding the description of the purpose of the study.

The authors cited literature and stated that in nonmedical education, studies showed that psychological safety influences students’ learning behaviors and performance, whereas, in clinical medical education, only a few studies demonstrated the effects of psychological safety on education. Readers of the Introduction section may assume that the purpose of the current study is “to determine how psychological safety affects education in the medical education area.”

However, the authors investigated factors influencing psychological safety in the clinical education field, rather than the impact of psychological safety on education.

1 Therefore, first, please clarify whether the purpose of this study is to identify factors influencing psychological safety in the clinical education field or to identify the impact of psychological safety on clinical education for medical students.

2 If the latter, consider whether the method is appropriate to achieve the purpose.

3 If it is the former, I would like you to describe a logical and scientific reason why the authors investigated the factors affecting psychological safety in clinical education, especially regarding supervision. For example, how about citing previous studies on the effects of supervision on psychological safety outside clinical education settings?

6. PLOS authors have the option to publish the peer review history of their article (what does this mean?). If published, this will include your full peer review and any attached files.

Reviewer #1: No

Reviewer #2: No

---

## [Author Response · Author response to Decision Letter 0]

10 Apr 2023

- Manuscript has been formatted according to these specifications

- In the “design and ethics” section we have written that (1) participants received written information about the study, i.e. had information about the study before providing consent, and (2) that they had to click “send” to submit their responses and they were informed that clicking “send” was considered consent, i.e. they gave a digital consent without a signature (to ensure anonymity). We would be happy to rewrite this section so that it fits your requirements but are unsure what needs to be changed. 

- We have attached the data files (“Dataset” and “Data codebook”) to our upload. 

Reviewers' comments:

Reviewer's Responses to Questions

Comments to the Author

1. Is the manuscript technically sound, and do the data support the conclusions?

Reviewer #1: Yes

Reviewer #2: Partly

2. Has the statistical analysis been performed appropriately and rigorously? 

Reviewer #1: Yes

Reviewer #2: Yes

3. Have the authors made all data underlying the findings in their manuscript fully available?

Reviewer #1: Yes

Reviewer #2: Yes

4. Is the manuscript presented in an intelligible fashion and written in standard English?

Reviewer #1: Yes

Reviewer #2: Yes

5. Review Comments to the Author

- Thank you to both reviewers for giving useful feedback that helped improve our article. 

Reviewer #1: 

- Are there any impact/issues on psychological safety among medical students? If yes, please provide the evidence in order to show that this study is important

o We have rewritten the introduction to make clearer what research has been done in the past and why this study is important. In particular, we have now stated that:

 «In medical students’ supervised encounters with patients, where education meets health care, there was an association between the students’ perception of learning and psychological safety [9]. 

 This indicates that promoting psychological safety in medical students’ supervised encounters with patients is important. However, we have found no studies looking at the association between psychological safety and medical students’ experiences with other aspects of supervised patient encounters. Such knowledge can be helpful in unravelling what can be done to promote psychological safety in this setting.»

o . We hope this answers reviewer 1s question. 

- Provide some explanation related to your variables.

o We have added several paragraphs in the methods section to have a more detailed description of which variables were included and what they entail. 

- What method that the researcher used in selecting the participants?

o We have revised the first sentence of the “design and ethics” section so that it now reads “This was a cross-sectional survey where European medical students were invited to complete an online questionnaire.” Together with the “participants, recruitment, and data collection” section, particularly the paragraph “To recruit participants, the online survey was distributed as web-links in e-mails and on social media by student representatives in national medical student associations in the International Federation of Medical Students Associations (IFMSA) between March and October 2020. Individual students were also encouraged to share the link with their peers.” We hope it is now clear that we did not select individual participants but invited all European medical students we could reach. 

- State the reliability result for Maastricht Clinical Teaching Questionnaire (MCTQ) in this study

o In the new paragraphs on variables we have now added that the MCTQ “subscales had Cronbach’s alphas ranging from 0.74 to 0.87 in this study.” We have also added the Cronbach alpha value for the psychological safety scale.

- Under Questionnaire and variables, you mentioned that there are 12 independent variables were used in this study. Please what are the 12 independent variables.

-o We have added several paragraphs in the methods section to have a more detailed description of which variables were included and what they entail. 

- For the respondents, ‘not male’ refer to female or what? Please explain it

o We have added the sentences “In post-graduate medical education, research suggest gender impacts trainees' perception of psychological safety [15]. Male was chosen as the comparator, as some studies suggest males feel more psychologically safe [15].” We hope this explains our choice and what it entails. 

Reviewer #2: The authors investigated the relationship between supervisors’ attributes and behaviors and students’ psychological safety in clinical educational settings, an area of interest to readers. A sufficient number of data have been analyzed robustly quantitatively, and the methods are clear.

However, I have concerns regarding the description of the purpose of the study.

The authors cited literature and stated that in nonmedical education, studies showed that psychological safety influences students’ learning behaviors and performance, whereas, in clinical medical education, only a few studies demonstrated the effects of psychological safety on education. Readers of the Introduction section may assume that the purpose of the current study is “to determine how psychological safety affects education in the medical education area.”

However, the authors investigated factors influencing psychological safety in the clinical education field, rather than the impact of psychological safety on education.

1 Therefore, first, please clarify whether the purpose of this study is to identify factors influencing psychological safety in the clinical education field or to identify the impact of psychological safety on clinical education for medical students.

2 If the latter, consider whether the method is appropriate to achieve the purpose.

3 If it is the former, I would like you to describe a logical and scientific reason why the authors investigated the factors affecting psychological safety in clinical education, especially regarding supervision. For example, how about citing previous studies on the effects of supervision on psychological safety outside clinical education settings?

- The reviewers’ comments made us realise that our introduction was not up to the wanted standard, lacking a clear explanation of why this study is relevant and novel, as well as exactly what was investigated. We hope our revisions has made this clear as we now argue why psychological safety is important to medical students’ learning in supervised patient encounters and we need to learn more about how psychological safety comes to exist in this context. 

---

## [Editor Report · Decision Letter 1]

14 Apr 2023

Psychological safety in European medical students’ last supervised patient encounter: A cross-sectional survey

PONE-D-22-23578R1

Dear Dr. Cathinka, 

We’re pleased to inform you that your manuscript has been judged scientifically suitable for publication and will be formally accepted for publication once it meets all outstanding technical requirements.

Kind regards,

Yaser Mohammed Al-Worafi

Academic Editor

PLOS ONE
---

## [Editor Report · Acceptance letter]

18 Apr 2023

PONE-D-22-23578R1 

Psychological safety in European medical students’ last supervised patient encounter: A cross-sectional survey 

Dear Dr. Thyness:

I'm pleased to inform you that your manuscript has been deemed suitable for publication in PLOS ONE. Congratulations! Your manuscript is now with our production department. 

Kind regards, 

on behalf of

Professor Yaser Mohammed Al-Worafi 

Academic Editor

PLOS ONE